# OpenReview forum: "RuleAdapter: Dynamic Rules for training Safety Reward Models in RLHF"
_ICML.cc/2025/Conference — ICML 2025 poster_

### Official Review · Reviewer_Af7f · 2025-03-11

**Overall Recommendation:** 3

**Summary:**

The paper introduces a dynamic method for selecting safety rules in RLHF. Instead of using a fixed set, it adaptively chooses 5 out of 100 rules for each prompt–response pair based on the score difference (discrepancy) and rule relevance. This approach is both theoretically justified and empirically validated, leading to improved safety performance.

**Claims And Evidence:**

The claims are clear and supported by both theoretical and empirical evidence.

**Essential References Not Discussed:**

The related works are properly cited.

**Experimental Designs Or Analyses:**

The main experimental results are strong, and the ablation studies are comprehensive. In Table 1, the authors compare the scores of RAMO with various baselines on RewardBench, and in Table 2, it is notable that RAMO outperforms the AllRules.

**Methods And Evaluation Criteria:**

- The motivation is clear, and the method is interesting. However, I have concerns regarding the generalization of the classification model. Did you hold out a validation set to evaluate its multi-label classification performance on unseen scenarios? What does accuracy look like?
- I also have some concerns regarding the motivation. While generating 100 rules can naturally lead to overlap and duplicates, this may not be the best practice. It might be more effective to carefully tune the prompts to generate only the 5 most critical and representative rules, which could be sufficient to achieve decent performance.
- Evaluating the reward model on safety benchmarks like RewardBench and SafetyBench provides a comprehensive view of the model's performance. These benchmarks are designed to assess various aspects of safety (e.g., refusals, helpfulness, and correctness), making them appropriate for this application.

**Other Comments Or Suggestions:**

I don't have additional comments. My comments and questions are in the above chunks.

**Other Strengths And Weaknesses:**

The paper creatively combines dynamic rule selection with multi-attribute reward modeling, extending prior work that relied on fixed or random rule selection. This dynamic approach, backed by theoretical analysis using information-theoretic concepts, provides a novel way to optimize preference labeling in RLHF.

**Questions For Authors:**

My questions and concerns are listed in Methods And Evaluation.

**Relation To Broader Scientific Literature:**

Prior work (e.g., Anthropic's Constitutional AI and OpenAI's 21 safety rules) typically applies a fixed set or randomly selected rules to label preferences. In contrast, this paper introduces a dynamic rule selection mechanism—selecting the top 5 rules based on maximum discrepancy and relevance for each prompt–response pair—which offers a more adaptive and informative approach. This idea refines earlier efforts in multi-attribute reward modeling by tailoring rule selection to the context at hand.

**Theoretical Claims:**

The theoretical claims are reasonable.

---

> ### Author Rebuttal · Authors · 2025-03-31
>
> We really appreciate the comments and compliments by Reviewer. Below we provide our responses to the concerns:
>
> # 1. Validation Performance of RuleAdapter during Training:
> We use a validation set of size 500 during the training of RuleAdapter. During training, the best checkpoint achieves: accuracy 0.967, precision 0.9645, and recall 0.9658 on the validation set.
>
>
> # 2. Duplication of 100 Rules:
> As mentioned in paper, we initially generated 400 raw safety rules. During generation, we considered the 19 safety categories from PKU and 133 from CollectiveCAI (see Section 4.1 for details and citations). Then we perform a **deduplication step** and reduce the pool to 100 rules. We have **manually checked** that our rule set almost covers all the safety aspects in PKU, CollectiveCAI, and other papers cited in Section 2, with no obvious duplication (some rules might talk about similar topics but they evaluate safety from different aspects). This confirms that our final set of 100 rules achieves a suitable balance of diversity and independence.
>
> # 3. Generate Only 5 Critical Rules
> If the reviewer means prompting strong LLMs such as GPT to dynamically generate 5 critical rules for each sample, in the Ablation Study (Section 5.1), we included this baseline and the averaged Safety performance is 92.8, not comparable to the SOTA performance (score 95.1) of our RAMO.
>
> For a fixed set of 5 rules, we considered baselines including random 5 rules, and fixed 5 rules selected randomly at the beginning. f the reviewer suggests selecting the 5 most critical and representative rules upfront and applying them across all data, we have conducted this additional experiment. These 5 critical rules were selected by GPT and verified by the authors. The results are as follows:
>
> |     Model   |DoNotAnswer|RefusalsDangerous|RefusalsOffensive|XstestShouldRefuse|XstestShouldRespond|Safety|
> |-------------|-----------|-----------------|-----------------|------------------|-------------------|------|
> |Critial5Rules|77.2       |94.0             |99.0             |96.1              |96.0               |92.7  |
>
> We see the **performance is not comparable to RAMO, and even slightly worse than using dynamic 5 rules by GPT**. Hence this further confirms the effectiveness of our dynamic-rule scheme based on the max-discrepancy strategy.
>
>
> We really thank the reviewer for all the comments and sincerely hope our clarification and additional experiments would address the concerns. Those explanations and additional experiments will be added to the paper, and we sincerely hope the reviewer could consider raising the score.

---

### Official Review · Reviewer_EEMq · 2025-03-12

**Overall Recommendation:** 3

**Summary:**

This paper introduces a dynamic approach to selecting safety rules for training reward models in RLHF. Rather than applying a fixed set of rules or randomly sampling from a large rule pool, the authors propose a method that adaptively selects the most critical rules for each pair of responses. Contributions:
- A framework that selects rules based on maximum discrepancy between paired responses
- Theoretical justification showing this strategy optimizes mutual information between rule-based labeling and ground-truth preferences
- An 8B parameter reward model that achieves top performance on the RewardBench safety benchmark

**Claims And Evidence:**

Claims about the supremacy of the method are partially supported by RAMO's performance on RewardBench. However, the theoretical claim that maximizing discrepancy optimizes mutual information, while mathematically derived, lacks empirical validation with real human preferences. The method is only evaluated on synthetic data or existing preference datasets, not with new human evaluations.

**Essential References Not Discussed:**

n/a

**Experimental Designs Or Analyses:**

The experimental design has several limitations:
- The preference datasets used are synthetic, with labels generated by LLMs rather than humans
- The effectiveness of the final aligned models is only evaluated on closed benchmarks, not real-world usage
- The 100-rule pool, while diverse, is still a constructed set and may not comprehensively cover all safety concerns
The ablation studies are reasonably thorough but don't fully explore whether the performance gains come from the dynamic rule selection or simply from using a more diverse set of rules during training.

**Methods And Evaluation Criteria:**

The method of dynamically selecting rules is reasonable for the problem of safety alignment. Using a maximum discrepancy approach to identify where responses differ most substantially is conceptually sound. However, the evaluation is limited to RewardBench, a single close-ended benchmark. There is no open-ended evaluation of the resulting models, which would better demonstrate real-world safety improvements. The lack of human evaluation is particularly problematic for safety claims, as automated benchmarks may not capture all nuances of human values.

**Other Comments Or Suggestions:**

n/a

**Other Strengths And Weaknesses:**

Strengths
- The framework for rule selection is well-developed, with theoretical grounding.
- The performance on RewardBench is impressive, especially for a model of its size.
- The integration into a full RLHF pipeline shows practical application potential

Weaknesses:
- The idea of using multiple rules is not a particularly novel invention, as similar approaches exist in multi-attribute reward models
- The evaluation is only on RewardBench, a single close-ended benchmark, with no open-ended evaluation.
- There is no human evaluation of the quality of the dynamically selected rules or the resulting model outputs.
- The approach still relies on large language models to evaluate against rules, which may propagate their biases.
- The paper doesn't fully address how to ensure the rule pool itself is comprehensive and unbiased.

**Questions For Authors:**

n/a

**Relation To Broader Scientific Literature:**

The paper builds on several research directions:
- multi-attribute reward modeling
- fine-grained annotation approaches in RLHF
- constitutional AI
The work extends these approaches by introducing dynamic rule selection. However, the core idea of using multiple facets or principles for evaluation is not fundamentally novel, as similar approaches appear in Constitutional AI and multi-attribute reward modeling. The main contribution is the adaptive selection mechanism rather than a new conceptual framework.

**Theoretical Claims:**

The proofs seem correct in broad strokes. I did not check the correctness of the proofs in detail.

---

> ### Author Rebuttal · Authors · 2025-03-31
>
> We sincerely appreciate the comments and provide our responses below (the new evaluation results will be added to the paper).
>
> # 1. Human Evaluation on Selected Rules:
> In Appendix F, we have provided **case studies with real data trios and analysis on the selected rules by RuleAdapter**. To further address the reviewer’s concern, we **conducted an evaluation** involving 8 volunteers (all are PhD or master students in STEM) to assess 100 randomly selected trios from HH-RLHF: volunteers are asked to provide binary labels indicating whether each rule selected by RuleAdapter is related and critical to the given trio. After averaging and scaling by the max score of 5, we obtain a **quality score of 89% (std 7%), meaning on average, 4.45 of the 5 rules are considered relevant and important, showing that RuleAdapter successfully selects out important rules dynamically**.
>
> # 2. Open-ended Evaluation of the Resulting Models:
> First we would like to emphasize that **RewardBench is a combined benchmark dataset containing multiple tasks**. For safety, it includes 5 datasets: DoNotAnswer, RefusalsDangerous, RefusalsOffensive, XTestShouldRefuse, XTestShouldRespond. These components make RewardBench a well-recognized, comprehensive, and robust benchmark for evaluating reward models. Our RAMO demonstrates SOTA performance on almost all safety tasks. Besides our evaluation of the reward model using RewardBench, we also evaluated the aligned model using SafetyBench, to ensure comprehensiveness.
>
> Second, to address the reviewer’s concern about open-ended human evaluation, we **conducted a survey involving 200 randomly selected prompts from HH-RLHF**. We collect responses from the original Llama3.2-1B-Intruct and the further aligned model guided by RAMO. For each pair, 5 human annotators compared and chose the response that is less harmful, less offensive, more ethical, and aligns better with safety guidelines. The **average win rate is 91% (std 7%), clearly indicating that RAMO effectively provides accurate rewards to enhance the model’s safety**.
>
> # 3. Labels Generated by LLMs instead of Human:
> One of our contributions is that using the fine-grained ratings based on **dynamic, critical rules simplifies and enhances the accuracy of the final rating compared to overall ratings, thus lowering accuracy requirements for the rater**. This advantage is validated by our SOTA performance in Table 1. Also, we highlight that human ratings are substantially more costly than our rater–the open-source Llama3 model. We **manually evaluated rule-based ratings on over 50 samples and found that the Llama3 ratings according to specific rules are quite reasonable, especially compared to giving an overall rating** for each sample (even humans can have huge disagreement in this case). We will provide more details about this part and include a few case studies in the revised paper.
>
> # 4. 100 Rules May Not Comprehensively Cover Safety Concerns:
> As mentioned in paper, we initially generated 400 raw safety rules. During generation, we considered the 19 safety categories from PKU and 75 public constitutions + 58 standard constitutions from CollectiveCAI (see Section 4.1 for details and citations). The **133 constitutions from CollectiveCAI are collected from humans and established documents**. Then we perform a deduplication and reduce the pool to 100 rules. We have manually checked that our rule set almost covers all the safety aspects in PKU, CollectiveCAI, and other papers cited in Section 2. This confirms our rule set’s comprehensiveness and effectiveness in representing critical safety aspects people care about.
>
> # 5. Is the Gain Simply from a More Diverse Rule Set?
> If this is the case, then possible baselines are: A. **Use all 100 rules**. B. Under the same setting of a 5-rule budget, **use 5 random rules**. C. **Human-selected 5 rules**. D. Similar to our dynamic scheme, but **query GPT to select 5 rules dynamically**. In Ablation Study (Table 2) and the additional experiment in the response to Reviewer Af7f, we have **compared against all these baselines**, and none of them demonstrate comparable performance** to our method. Moreover, querying GPT entails a higher cost compared to our RuleAdapter (a Llama3.2-3B model). Hence the ablation study confirms that the performance gain is not simply from a more diverse set of rules.
>
> # 6. Novelty of Multi-principle Evaluation:
> We **did not claim the novelty of using multi-principle evaluation in our contribution list**. Our main novelty is the dynamic rule scheme for the fine-grained rule-based rating, and we use comprehensive experiments, theoretical justification, and thorough ablation studies to show the effectiveness of our framework.
>
> We deeply appreciate the reviewer’s insightful comments and valuable suggestions. We sincerely hope our clarification and additional evaluations above would address the concerns, and respectfully hope the reviewer could raise the score.

---

### Official Review · Reviewer_A2RT · 2025-03-14

**Overall Recommendation:** 3

**Summary:**

The paper introduces a dynamic approach to RLHF that adaptively selects the most critical rules for evaluating response pairs, moving beyond traditional binary preference selection. The authors provide mathematical justification showing their method optimizes mutual information between rule-based labeling and ground-truth preferences. Their 8B reward model reportedly achieved the highest safety performance on RewardBench, outperforming larger models. The work addresses key challenges in fine-grained annotation approaches where human opinions vary and comprehensive response comparison is difficult.

**Claims And Evidence:**

The dynamic selection of rule is important.
Exist approach (e.g., randomly select rule, applying a large number of rules, using a small fixed set of rules) is sub-optimal.

**Essential References Not Discussed:**

NA

**Experimental Designs Or Analyses:**

The use GPT-4 to generate 400 raw safety rules and finally select 100 rules.
To train rule adapter, they selected 5K prompts from ShareGPT (a dataset featuring real user conversations). The corresponding responses are generated using 6 models (at 7B scale). The rule adapter is trained as a multi-label classification task.
The critical rule is identified by the max-discrepancy strategy proposed in section 3.3

**Methods And Evaluation Criteria:**

The propose dynamic rule selection is motivated by the fact: during reward model training, it relies on the trio and the preference label to learn the preference of chosen and rejected responses.

The paper develops a rule selection strategy based on the max discrepancy measure and train the Rule Adapter to achieve the dynamic selection of the most critical rules, enhancing the quality and interpretability of preference labeling.

**Other Comments Or Suggestions:**

NA

**Other Strengths And Weaknesses:**

The accuracy of the max-discrepancy strategy itself is very important, which need more details explanation and support by experiments

**Questions For Authors:**

NA

**Relation To Broader Scientific Literature:**

The rule adapter is important for fully automatic RLAIF without human preference labeling.

**Theoretical Claims:**

The paper theoretically prove that our max-discrepancy method effectively maximizes the mutual information between the preference labels by the selected rules and the hidden ground-truth preference labels.

---

> ### Author Rebuttal · Authors · 2025-03-31
>
> We really appreciate the comments. For the request for more details on the accuracy of the max-discrepancy strategy, since “accuracy of the max-discrepancy strategy” can be interpreted in several ways, below we provide our responses to all the interpretations:
>
> # 1. Validation Performance of RuleAdapter during Training:
> We use a validation set of size 500 during the training of RuleAdapter. During training, the best checkpoint achieves: accuracy 0.967, precision 0.9645, and recall 0.9658 on the validation set.
>
> # 2. Quality of Rules Selected by RuleAdapter:
> In Appendix F, we have provided **case studies with real data trios and analysis on the selected rules by RuleAdapter**. To further evaluate the quality of selected rules, we  **conducted an evaluation survey** involving 8 volunteers (all are PhD or master students in STEM) to assess 100 randomly selected trios from HH-RLHF: volunteers are asked to provide binary labels indicating whether each rule selected by RuleAdapter is related and critical to the given trio. After averaging and scaling by the max score of 5, we obtain a **quality score of 89% (std 7%), meaning on average, 4.45 of the 5 rules are considered relevant and important, showing that RuleAdapter successfully selects out important rules dynamically**.
>
> # 3. Accuracy of the Aligned Models:
> First, RewardBench itself evaluates the performance of reward models by their accuracy on the binary preferences on 5 preference datasets: DoNotAnswer, RefusalsDangerous, RefusalsOffensive, XTestShouldRefuse, XTestShouldRespond. These components make RewardBench a well-recognized, comprehensive, and robust benchmark for evaluating reward models. Our RAMO demonstrates SOTA performance on almost all safety tasks. For example, on RefusalsOffensive, RAMO achieves a score 99.0, meaning that for 99% of the test data, RAMO matches human’s annotated preferences. Besides our evaluation of the reward model using RewardBench, we also evaluated the aligned model using SafetyBench, to ensure comprehensiveness.
>
> Second, to further provide open-ended human evaluation, we **conducted a survey involving 200 randomly selected prompts from HH-RLHF**. We collect responses from the original Llama3.2-1B-Intruct and the further aligned model guided by RAMO. For each pair, 5 human annotators compared and chose the response that is less harmful, less offensive, more ethical, and aligns better with safety guidelines. The **average win rate is 91% (std 7%), clearly indicating that RAMO effectively provides accurate rewards to enhance the model’s safety**.
>
> We thank the reviewer for all the comments and sincerely hope our clarification and additional evaluations above would address the concerns. These details and explanations will be added to the paper, and we respectfully hope the reviewer could consider raising the score.

---

### Decision · Program_Chairs · 2025-05-01

**Decision:**

Accept (poster)

**Comment:**

This work focuses on training safety-specific reward models, by (a) generating a somewhat large (100) set of rules to be used when deciding which response to prefer, (b) proposing a dynamic selection scheme to select a small (5) subset of rules to apply to each pair of responses to rank, and (c) training a BT reward model on the resulting dataset of preferences. The empirical validation, focused on the safety subset of RewardBench, shows the SotA performance of the approach.

Following the discussion period, all reviewers are leaning towards acceptance, though not being particularly enthusiastic (all 3's). I personally find the empirical results to be convincing enough to be worth sharing at ICML, especially as safety is a primary concern in LLM training. However, as acknowledged by the authors in the their conclusion and also mentioned by some reviewers, I find the proposed approach a bit "ad-hoc" as it's hard to be convinced that generated/selected rules are optimal (both in terms of quantity and content). Still, I expect safety researchers to be interested in those findings and in the pool of rules resulting from this work, so I would also lean towards acceptance.